# Body Positivity and Eating Behaviors Among Women Attending Fitness Classes: Associations with Body Mass Index

**DOI:** 10.3390/healthcare13233122

**Published:** 2025-12-01

**Authors:** Martyna Kłoda, Monika Marszołek, Wiktoria Staśkiewicz-Bartecka, Sylwia Jaruga-Sękowska, Małgorzata Magdalena Michalczyk

**Affiliations:** 1Department of Physical Activity and Health Promotion, The Jerzy Kukuczka Academy of Physical Education in Katowice, 40-065 Katowice, Poland; martynakloda.dietetyka@gmail.com (M.K.); m.marszolek@awf.katowice.pl (M.M.); 2Department of Food Technology and Quality Evaluation, Department of Dietetics, Faculty of Public Health in Bytom, Medical University of Silesia in Katowice, ul. Jordana 19, 41-808 Zabrze, Poland; wstaskiewicz@sum.edu.pl; 3Department of Health Promotion, Faculty of Public Health in Bytom, Medical University of Silesia in Katowice, ul. Piekarska 18, 41-902 Bytom, Poland; 4Institute of Sport Science, The Jerzy Kukuczka Academy of Physical Education in Katowice, 40-065 Katowice, Poland; m.michalczyk@awf.katowice.pl; 5Medict Institute, ul. Rolników 142, 44-141 Gliwice, Poland

**Keywords:** body image, emotional eating, cognitive restraint, fitness, women, self-acceptance

## Abstract

**Background**: Body positivity is a social movement aimed at promoting acceptance and appreciation of diverse body types. Despite its growing popularity, its relationship with eating behaviors and body mass index (BMI) remains unclear. This study aimed to investigate the opinions of women attending fitness classes on the body positivity movement and to assess the association between these attitudes, eating behaviors, and BMI. **Methods**: A total of 118 women aged 18–65 years participated in the study. Data were collected using a self-developed body positivity questionnaire and the validated Polish version of the Three-Factor Eating Questionnaire (TFEQ-13). Participants were stratified into two groups based on BMI (<25 and ≥25 kg/m^2^). **Results**: No significant association was found between general attitudes toward body positivity and either eating behaviors or BMI. However, a positive correlation was observed between BMI and emotional eating as well as cognitive restraint. Women with higher BMI demonstrated greater cognitive restraint and tendencies toward emotional eating. Body acceptance was significantly lower in women with BMIs ≥ 25. **Conclusions**: While general attitudes toward the body positivity movement do not appear to influence eating behaviors directly, body weight is linked to both emotional eating and cognitive restraint. These findings underscore the complex relationship between psychological factors, eating patterns, and body image, suggesting the need for further research and tailored interventions promoting both health and self-acceptance.

## 1. Introduction

The body positivity movement promotes a positive attitude towards one’s own body, self-acceptance, and counteracting discrimination based on appearance. Its origins date back to 1996, when Connie Sobczak and Elizabeth Scott founded The Body Positive organization, whose goal was to cultivate a healthy attitude towards one’s own body and strengthen self-acceptance [1]. In recent years, the body positivity movement has gained popularity thanks to social media, which exposes images and narratives promoting body diversity and positive body image. Research shows that exposure to such content can improve body image and self-esteem [2,3]. At the same time, there are concerns that excessive acceptance of one’s appearance in the case of overweight or obese people may reduce motivation to change lifestyle and take health-promoting actions, such as weight control or physical activity [4].

Obesity is one of the biggest public health problems, and its prevalence is growing worldwide. According to World Obesity Atlas projections, by 2030, the percentage of overweight and obese people could reach as high as 50% of the population [5]. This problem also affects children—between 1975 and 2025, the number of obese school-age children increased from 11 million to 124 million [6]. Obesity is a multifactorial phenomenon, and its development is associated not only with diet and physical activity levels, but also with eating disorders and psychological and social factors. These disorders include emotional eating, uncontrolled food consumption, and cognitive restraint, which can contribute to weight gain and make it difficult to lose weight [7,8,9]. There is also a strong link between the stigmatization of overweight or obese people and unhealthy eating patterns, both in the form of external discrimination and self-stigmatization. People who experience negative perceptions of their body weight are more likely to overeat and engage in emotional eating, which promotes weight gain and hinders Body Mass Index (BMI) control [10].

Considering both the potential benefits and risks associated with the body positivity movement, it is important to examine how attitudes toward body positivity influence eating behaviors and body mass indices. The existing literature indicates that exposure to body positivity content improves self-esteem and body image [2,4], but may also reduce the desire to be thin among overweight individuals [3,11], raising questions about the possible health consequences of such acceptance. In addition, interventions based on self-compassion, which are consistent with the philosophy of body positivity, have shown improvements in eating behaviors, physical activity, and weight loss in some studies [8,12].

In the context of the impact of stigma on eating behaviors, it is critical to consider whether the body positivity movement influences eating patterns. That is, does greater body self-acceptance promote healthier eating habits? The current literature does not offer a clear answer to this question. Based on previous research indicating that the body positivity movement can affect body image, self-acceptance, and psychological well-being, albeit with limited evidence on its relationship with eating habits and BMI, we hypothesized whether women with a higher BMI (≥25) would express more critical opinions about the body positivity movement and report higher levels of emotional eating and cognitive restraint compared to women with a normal BMI. Therefore, to address this deficiency, the present study evaluated the opinions of women participating in fitness classes about the body positivity movement and the relationship between these opinions, eating habits, and BMI.

Since existing psychometric tools did not allow for a comprehensive assessment of attitudes toward body positivity in relation to eating behaviors and body mass indices, we developed an original questionnaire for this cross-sectional study that would allow for the simultaneous analysis of participants’ opinions, eating behaviors, and BMI scores at a specific time point. The results would provide a better understanding of whether promoting self-acceptance through body positivity contributes to greater well-being and a healthy lifestyle, or whether, under certain circumstances, it can limit motivation to change pro-health behaviors.

## 2. Materials and Methods

### 2.1. Procedure of the Study

This qualitative and cross-sectional study was conducted between February and June 2025 using a mixed-method approach. Data were collected using the computer-assisted web interview (CAWI) method with an online questionnaire (participants were recruited through posts published on gym websites and social media), supplemented by face-to-face surveys conducted in fitness clubs. This method is considered accurate and reliable in psychological research. Participants were given detailed information about the objectives of the study, assured of anonymity, and provided with instructions on how to complete the questionnaire. All research procedures were reviewed and approved by the Bioethics Committee of the Medical University of Silesia in Katowice (protocol code BNW/NWN/0052/KB/229/23) and were conducted in accordance with the principles of the Declaration of Helsinki. Participants were informed about the voluntary nature of participation, the right to withdraw at any time without consequences, and the confidentiality and anonymity of their responses.

### 2.2. Participants

The inclusion criteria were: (1) voluntary participation and complete questionnaire completion; (2) female sex; (3) age over 18 years; and (4) regular physical activity, defined as participation in at least three training sessions per week at a fitness club. The exclusion criteria included: (1) lack of regular physical activity; (2) self-reported chronic conditions (e.g., metabolic disorders) that could significantly affect eating behaviors, body composition, or body image; and (3) pregnancy, which could interfere with the relationship between body image, eating behaviors, and BMI.

### 2.3. Survey Tools

The survey questionnaire consisted of a demographic section covering sociodemographic data (age, height, weight, presence of chronic diseases, education, physical activity) and two research tools: a validated Polish version of the Three-Factor Eating Questionnaire (TFEQ-13) [13] and an original questionnaire examining opinions on the body positivity movement. The TFEQ-13 is a validated Polish version that assesses three dimensions of eating behaviors: cognitive restraint, emotional eating, and uncontrolled eating. Original questionnaire on body positivity: Designed to assess participants’ opinions and attitudes towards the body positivity movement [14].

Responses to the TFEQ-13 are recorded on a 4-point Likert scale, with options ranging from “Strongly agree” to “Strongly disagree,” allowing participants to indicate the extent to which each statement applies to them. The emotional eating subscale measures the tendency to overeat in response to negative emotions or stress, with a maximum score of 9 points. The uncontrolled eating subscale assesses the risk of excessive food consumption due to impaired self-regulation and feelings of uncontrollable hunger, with a maximum score of 12 points. The cognitive restraint subscale evaluates the deliberate restriction of food intake to control body weight and body image, with a maximum score of 12 points. Each subscale is scored separately, with higher scores indicating a greater degree of the eating behavior in question [13,14].

An original questionnaire consisting of 12 questions was developed to assess participants’ opinions on positive body image. Responses were structured using a Likert scale with five options: “Definitely yes,” “Probably yes,” “I don’t know,” “Probably no,” and “Definitely no.” The answers were coded on a scale from 0 to 4, and the results were converted into points, with higher values indicating a more critical attitude towards the body positivity movement and lower values indicating a more positive attitude. The maximum possible score was 48 points. The reliability of the tool was assessed using Cronbach’s α coefficient, which was 0.86, indicating satisfactory internal consistency. However, it should be noted that the lack of full psychometric validation limits the interpretation of the results and requires caution when drawing conclusions based on them (Table 1).

### 2.4. Statistical Analyses

Statistical analyses were performed using Statistica v.13.3 software (StatSoft Polska Sp. z o.o., Kraków, Poland). Quantitative variables were expressed as means and standard deviations (M ± SD), while qualitative variables were presented as absolute numbers and percentages. The normality of distributions was verified with the Shapiro–Wilk test. For comparisons between two groups, the Mann–Whitney U test was applied for continuous variables not meeting the assumption of normality, and the chi-square (χ^2^) test of independence was used for categorical variables. Effect sizes for categorical data were evaluated using Cramer’s V coefficient. In addition, linear regression analyses were performed to identify predictors of BMI and age based on TFEQ-13 subscale scores (emotional eating, uncontrolled eating, and cognitive restraint). Regression coefficients (B), standard errors (SE), *t* statistics, and *p* values were reported. The level of statistical significance was set at α = 0.05.

## 3. Results

A total of 118 women aged 18–65 participated in the study. Among them, 67 had a BMI < 25 kg/m^2^ and 51 had a BMI ≥ 25 kg/m^2^. The characteristics of the study group are presented in Table 2. The lowest BMI among participants was 17.9 kg/m^2^, and the highest was 34.9 kg/m^2^. A statistically significant difference between the two groups was observed in age, height, body mass, and BMI.

### 3.1. Attitudes Towards Body Positivity

The arithmetic mean of body positivity opinion scores did not significantly differ between women with a BMI above 25 kg/m^2^ and those below this value (Table 3).

Meanwhile, responses to several questions revealed a substantial difference between the groups (Table 4). Statistically significant differences were observed for questions 5, 6, 7, 9, and 13. Additionally, the difference in responses within the group for question 2 approached statistical significance.

### 3.2. Body Mass Index, Eating Behaviors, and Opinion About Body Positivity

The average scores of women participating in the study on the TFEQ-13 subscales are presented in Table 5. In all participants, the mean score on the cognitive restriction subscale was higher than on the uncontrolled eating and emotional eating subscales. Women with a BMI > 25 scored significantly higher on the cognitive restraint subscale compared to women with a normal BMI. However, the differences observed in emotional eating and uncontrolled eating between BMI groups did not reach statistical significance.

The analysis revealed that emotional eating and cognitive restraint were statistically significant predictors of BMI, whereas uncontrolled eating was not significant (Table 6). The model accounted for only a small proportion of the variance.

Two significant predictors were identified: emotional eating and cognitive restraint (Table 7). Emotional eating was negatively associated with age. Cognitive restraint showed a positive association with age. Uncontrolled eating was not significantly associated with age. The model explained only 7.2% of the variance in age.

## 4. Discussion

The primary objective of this study was to provide a better understanding of whether promoting self-acceptance through body positivity contributes to greater well-being and a healthy lifestyle, or whether, under certain circumstances, it may limit motivation to engage in health-promoting behaviors. The results of the study show the complex nature of the relationship between perceptions of the body positivity movement, body weight, and eating behaviors among women participating in fitness classes. The lack of a direct relationship between the general attitude towards body positivity and BMI and eating behaviors is consistent with the findings of Paulisova and Orosova, which indicate that body dissatisfaction is shaped by a complex interaction of sociocultural factors, self-objectification mechanisms, and processes related to individual autonomy. This may suggest that merely declaring acceptance of the idea of body positivity is not enough to influence daily health habits if the individual continues to experience the pressure of social norms and comparisons [15].

### 4.1. Body Acceptance

The results suggest that lower body acceptance in women with a BMI ≥ 25 is associated with specific eating behaviors. Higher levels of emotional eating correlate with a higher BMI. Additionally, the positive relationship between cognitive restriction and BMI indicates that individuals who consciously attempt to control their food intake often have difficulty maintaining their weight effectively. This mechanism can be interpreted in light of the emotional eating model proposed by van Strien et al., according to which negative affect and depressive symptoms promote weight gain through an increased tendency to reach for food in response to emotions [16]. Numerous studies have confirmed that overweight individuals are more likely to report using emotional eating as a mood regulation strategy [17,18,19]. Importantly, this mechanism is not limited to short-term relief of emotional tension, but in the long term contributes to the perpetuation of overweight and obesity [20]. A higher level of cognitive restriction may indicate compensatory attempts to control body weight. Löffler et al. [21] showed that people with elevated BMI more often report using conscious strategies to restrict eating. However, these strategies do not always lead to effective weight loss, especially when emotional eating is also present [22].

Furthermore, research by Staśkiewicz-Bartecka et al. [23] indicates that body dissatisfaction and the accompanying negative emotions increase the risk of compensatory behaviors, including overeating or consuming diet and low-calorie products in ways that do not meet the body’s actual needs. Brenton-Peters et al. drew similar conclusions, emphasizing that low body acceptance and lack of self-compassion can limit the effectiveness of dietary interventions, while programs that address both of these components support not only weight loss but also improved quality of life [8]. These relationships are consistent with the observations of Wójcik, who showed that dissatisfaction with one’s own body reduces both life and sexual satisfaction, thus affecting key aspects of quality of life [24].

### 4.2. Age and Eating Behaviors

The study revealed certain links between age and eating behaviors, although these explain only a small part of the variability in the results. The intensity of emotional eating decreases with age, which may be due to better coping with stress and emotions. In turn, the increase in cognitive restraint among older participants may indicate greater awareness and control over food intake. Similar observations were made in studies conducted by Alqahtani and Alhazmi, who found that among medical professionals, older age was associated with higher levels of cognitive restraint and lower levels of emotional eating [25]. However, as the authors note, sociodemographic factors such as educational attainment and occupational status may also influence these relationships [25].

It is also worth noting that other studies, such as a meta-analysis conducted by Zhou and colleagues, indicate a stronger tendency toward emotional eating among younger individuals, especially women, which may suggest differences in emotion regulation mechanisms depending on age and gender. However, as both Alqahtani and Alhazmi and Zhou emphasize, further research is needed to better understand the mechanisms underlying these relationships and their potential implications for health interventions [25,26].

### 4.3. Opinions Regarding Media Messages

The results of our study indicate that participants with higher BMI values were more likely to express more polarized opinions regarding media messages related to body positivity. This may reflect greater sensitivity to social messages about body appearance, which is consistent with previous observations indicating that individuals with higher BMIs are more likely to experience weight stigma and internalize social ideals of thinness [27]. These results are consistent with the findings of Izydorczyk et al., who showed that the internalization of the athletic ideal mediated the influence of family and body image among young women in different cultural contexts [28].

The cross-cultural perspective shows that body positivity functions differently depending on the sociocultural context. Hanson et al. showed that women from Nigeria have the highest level of body appreciation, followed by Chinese women, while the lowest values were recorded in Western countries, which was associated with the level of internalization of the ideal of a slim figure and social and media pressure [29]. Similar differences are described in a systematic review of body image research, indicating that in Western cultures, the dominance of the ideal of thinness contributes to increased body dysphoria, while in other contexts, social and cultural factors such as status and age play a greater role [30]. In turn, research from China suggests that a positive relationship with one’s own body acts as a protective factor against symptoms of depression and anxiety [31]. In Poland, as in other Western countries, body positivity appears to function primarily as a point of reference and as a topic of debate on health and appearance. In our sample of physically active women, body positivity did not directly correspond to changes in daily behaviors, consistent with findings from international studies.

The role of social media in shaping contemporary standards of beauty and content related to the body positivity movement is significant. Cohen, Newton-John et al. report that exposure to content promoting body diversity can improve body image and self-esteem [32]. Previous studies have also shown that exposure to “fitspiration” content can increase social comparisons and body dissatisfaction, particularly among physically active women [33,34]. In our sample, participants reported ambivalent experiences related to physical activity. These findings are consistent with earlier research indicating that a positive body image can enhance self-esteem and mental well-being, although it does not always result in changes in health-related behaviors [2,4,32].

In addition to body positivity, other emerging concepts such as body neutrality and fitspiration provide important insights into contemporary discourse on body image. Body neutrality shifts the focus from appearance to functionality and acceptance, offering an alternative framework that can reduce the pressure associated with physical ideals [35]. Furthermore, recent experimental findings suggest that body-neutral content on social media may be more effective in protecting against negative mood and body dissatisfaction than weight-stigmatizing content or even body-positive messages [36]. On the other hand, exposure to fitspiration content has been shown to lead to unfavorable comparisons and decreased body satisfaction [37], although the inclusion of elements such as self-compassion may mitigate these negative effects [38]. Taken together, these perspectives highlight the complexity of online narratives about the body and suggest that future interventions should consider the differential effects of body positivity, body neutrality, and physical inspiration on body image and health-related behaviors.

In summary, the results complement the existing literature, indicating that attitudes toward the body positivity movement among women who exercise are largely dependent on BMI and perceived social and media pressures. This confirms the need for further research on how body positivity can be adapted in health practice—not as an idea opposed to health concerns, but as a tool that strengthens self-acceptance and supports healthy behaviors.

The findings may have many practical applications, especially in the context of promoting healthy lifestyles and psychological support for physically active women. Understanding how factors such as body weight or attitudes toward body positivity influence eating behavior allows for a more effective design of education programs. This is particularly relevant in the context of the growing problem of overweight and obesity, which is affecting a growing proportion of the population, including women. Excess body weight is often associated with reduced self-esteem and unhealthy eating patterns, which can be exacerbated by negative body image.

Equally important is the role of the findings in the context of a worsening mental health crisis, especially among women, who are increasingly experiencing symptoms of depression and anxiety disorders. Low self-esteem, body dissatisfaction, and social pressures related to appearance may be factors that exacerbate these problems. Moreover, identifying the link between attitudes toward body positivity and eating behaviors may inspire social campaigns and educational programs that promote positive attitudes toward one’s body, healthy eating, and physical activity as elements of well-being, rather than merely means to an ideal appearance. Finally, the findings can serve as a starting point for further research on psychological and behavioral aspects of health, helping to better understand the mechanisms underlying today’s health challenges.

### 4.4. Strengths and Limitations

The study has several important strengths. First of all, it included a fairly broad group of women between the ages of 18 and 65 who attend fitness clubs, which made it possible to analyze relationships across age groups and body composition. The well-validated and widely used TFEQ-13 was used to measure eating behavior, which increases the reliability of the results. In addition, data collection was performed both in person and via an online survey, which may have contributed to increased accessibility and participation of the respondents. In addition, the use of a variety of statistical analysis methods, such as Spearman correlations and linear regression, allowed for a comprehensive examination of the relationship between BMI, eating behavior, and attitudes toward body positivity movement.

However, the study also has several limitations. First, its cross-sectional design prevents drawing conclusions about causal relationships or changes over time. Additionally, weight and height data, as well as questionnaire responses, were self-reported, which introduces the risk of inaccuracies due to social desirability bias or memory errors. The study sample included only women who attend fitness clubs, limiting the generalizability of the findings to the broader population, especially men or individuals who are not physically active. It is also important to note that the study assessed attitudes toward the body positivity movement rather than body image itself, which may explain the absence of significant relationships between these attitudes and eating behaviors. Potential confounding variables such as psychological state, socioeconomic status, or cultural factors were not taken into account, which could have influenced the results.

## 5. Conclusions

The study showed that the overall attitude of women participating in fitness classes towards body positivity is not directly related to their eating behaviors or BMI. At the same time, a higher BMI was associated with both greater cognitive restriction and a tendency toward emotional eating. These results suggest that simply declaring a positive attitude towards one’s body does not necessarily translate into healthy eating habits, and interventions promoting self-acceptance should also take into account psychological and behavioral aspects such as emotional eating and food intake control. Promoting body positivity in the context of health and physical activity can support self-acceptance, but it requires the simultaneous promotion of healthy eating behaviors, especially in groups of people with higher BMIs.

## Figures and Tables

**Table 1 healthcare-13-03122-t001:** Questions included in the original questionnaire.

Questions	
1	Do you think that the concept of body positivity has a positive impact on the well-being of overweight people?
2	Do you think that the portrayal of overweight people in the media (advertisements, films) has a positive impact on the well-being of overweight or obese people?
3	Do you think that the portrayal of overweight people in the media (advertisements, films) makes it easier for overweight or obese people to accept their own bodies?
4	Do you think that the concept of body positivity supports people who are losing weight?
5	Do you think that the concept of body positivity helps overweight people to take a rational approach to dieting?
6	Do you think that the functionality of the body is more important than its appearance?
7	Do you accept your body?
8	In your opinion, does body positivity in the media promote obesity?
9	Do you think that the concept of body positivity presented in the media motivates overweight or obese people to lose weight?
10	Do you think that the concept of body positivity has an impact on reducing emotional eating among overweight people?
11	Do you think that the concept of body positivity makes it easier for overweight people to control the amount of food they eat?
12	Do you think that the concept of body positivity influences overweight people to reduce the amount of food they eat?

**Table 2 healthcare-13-03122-t002:** Characteristics of the group.

	Total (n = 118)	<25 (n = 67)	>25 (n = 51)	*p*-Value
Age [years] (X ± SD)	34 ± 11.7	31 ± 9.5	38.3 ± 13.0	<0.001 *
Min–Max	18–65	18–63	19–65	
Height [cm] (X ± SD)	167.6 ± 6.7	169.5 ± 6.2	165.2 ± 6.7	<0.001 *
Min–Max	145–185	157–185	145–183	
Body mass [kg] (X ± SD)	69.5 ± 11.3	62.2 ± 7	79 ± 8.5	<0.001 *
Min–Max	50–96	45–82	62–96	
BMI [kg/m^2^] (X ± SD)	24.8 ± 4.4	21.7 ± 1.9	29.6 ± 3.1	<0.001 *
Min–Max	17.9–34.9	17.9–24.9	25–34.9	

<25—respondents with a BMI of less than 25 kg/m^2^; ≥25—respondents with a BMI of 25 kg/m^2^ or higher; *p* < 0.05; X—average; SD—standard deviation, * *p* < 0.05.

**Table 3 healthcare-13-03122-t003:** Opinion about body positivity.

Variable	Total (n = 118) (X ± SD)	Min–Max	<25 (n = 67) (X ± SD)	Min–Max	>25 (n = 51) (X ± SD)	Min–Max	*p*-Value
Opinion about body positivity	23.5 ± 8.5	5–48	23.5 ± 7.6	5–42	23.6 ± 9.6	6–48	0.97

<25—respondents with a BMI of less than 25 kg/m^2^; ≥25—respondents with a BMI of 25 kg/m^2^ or higher; *p* < 0.05; X—average; SD—standard deviation.

**Table 4 healthcare-13-03122-t004:** Variation in participants’ responses to the specific questionnaire item.

	Total (n = 118)	<25 (n = 67)	>25 (n = 51)	*p*-Value	V-Cramer
Question 5. Does the concept of body positivity promote a rational approach to diet among people who are overweight?					
Definitely yes	32 (27.1%)	16 (23.9%)	16 (31.4%)	0.018	0.32
Probably yes	17 (14.4%)	16 (23.9%)	1 (2%)		
Probably no	41 (34.7%)	21 (31.3%)	20 (39.2%)		
Definitely no	16 (13.6%)	9 (13.4%)	7 (13.7%)		
I don’t know	12 (10.2%)	5 (7.5%)	7 (10.2%)		
Question 6. Do you think the concept of body positivity helps people with excessive body weight take a rational approach to dieting?					
Definitely yes	42 (35.6%)	21 (31.3%)	21 (41.2%)	0.038	0.293
Probably yes	10 (8.5%)	8 (11.9%)	2 (3.9%)		
Probably no	7 (5.9%)	1 (1.5%)	6 (11.8%)		
Definitely no	2 (1.7%)	2 (3%)	0 (0%)		
I don’t know	57 (48.3%)	35 (52.2%)	22 (43.1%)		
Question 7. Do you think that body functionality is more important than its appearance?					
Definitely yes	61 (51.7%)	32 (47.8%)	29 (56.9%)	<0.001	0.412
Probably yes	61 (51.7%)	32 (47.8%)	29 (56.9%)		
Probably no	23 (19.5%)	10 (14.9%)	13 (25.5%)		
Definitely no	5 (4.2%)	0 (0%)	5 (9.8%)		
I don’t know	27 (22.9%)	24 (35.8%)	3 (5.9%)		
Question 9. Does the body positivity message present in the media motivate overweight or obese people to take action aimed at losing weight?					
Definitely yes	22 (18.6%)	6 (9%)	16 (31.4%)	0.002	0.382
Probably yes	25 (21.2%)	19 (28.3%)	6 (11.8%)		
Probably no	50 (42.4%)	32 (47.8%)	18 (35.3%)		
Definitely no	18 (15.3%)	7 (10.4%)	11 (21.5%)		
I don’t know	3 (2.5%)	3 (4.5%)	0 (0%)		
Question 12. Do you think the way body positivity is portrayed in the media contributes to an increase in the number of people who are overweight or obese?					
Definitely yes	44 (37.3%)	22 (32.8%)	22 (43.1%)	0.032	0.300
Probably yes	23 (19.5%)	19 (28.4%)	4 (7.8%)		
Probably no	27 (22.9%)	14 (20.9%)	13 (25.5%)		
Definitely no	10 (8.5%)	7 (10.4%)	3 (5.9%)		
I don’t know	14 (11.8%)	5 (7.5%)	9 (17.6%)		

<25—respondents with a BMI of less than 25 kg/m^2^; ≥25—respondents with a BMI of 25 kg/m^2^ or higher; *p* < 0.05.

**Table 5 healthcare-13-03122-t005:** Results of questionnaire TFEQ-13.

Variable	Total (n = 118) (X ± SD)	Min–Max	<25 (n = 67) (X ± SD)	Min–Max	>25 (n = 51) (X ± SD)	Min–Max	*p*-Value
Emotional eating	3.1 ± 2.3	0–9	2.7 ± 2.0	0–9	3.6 ± 2.6	0–9	0.1
Uncontrolled eating	5.4 ± 3.2	0–15	4.8 ± 2.7	0–12	6.3 ± 3.7	0–15	0.06
Cognitive restraint	7.7 ± 3.4	0–14	7.1 ± 3.6	0–14	8.7 ± 2.9	2–14	0.01

<25—respondents with a BMI of less than 25 kg/m^2^; ≥25—respondents with a BMI of 25 kg/m^2^ or higher; *p* < 0.05; X—average; SD—standard deviation.

**Table 6 healthcare-13-03122-t006:** Linear regression analysis of TEFQ-13 score and BMI values (n = 118).

Predictor	Factor	SE	*t*	*p*
Emotional eating	0.48	0.23	1.95	0.04 *
Uncontrolled eating	0.12	0.16	0.74	0.46
Cognitive restraint	0.22	0.12	1.83	0.01 *

TFEQ-13—Three-Factor Eating Questionnaire; SE—standard error; *t*—*t*-statistics; * *p* < 0.05.

**Table 7 healthcare-13-03122-t007:** Linear regression analysis of TEFQ-13 score and age (n = 118).

Model Coefficients—TEFQ-13; R = 0.27 R^2^ = 0.072				
Predictor	Factor	SE	*t*	*p*
Emotional eating	−1.37	0.61	−2.25	0.027 *
Uncontrolled eating	0.32	0.42	0.758	0.45
Cognitive restraint	0.71	0.32	2.22	0.029 *

TFEQ-13—Three-Factor Eating Questionnaire; SE—standard error; *t*—*t*-statistics; * *p* < 0.05.

## Data Availability

The raw data supporting the conclusions of this article will be made available by the authors upon request due to privacy.

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
