# Peer review of "Body Positivity and Eating Behaviors Among Women Attending Fitness Classes: Associations with Body Mass Index"

_healthcare, 2025, doi:10.3390/healthcare13233122_

Round 1
Reviewer 1 Report
Comments and Suggestions for Authors
The authors of the article attempt to explain the benefits of body positivity regarding certain eating behaviors; however, the theoretical approach is weak in terms of demonstrating why women who attend fitness classes can assess these complex aspects using a single test. In the end, they do not find significant associations that contribute to pathological conditions related to these behaviors. It does not have a greater methodological support in order to explain the objectives that they expected to achieve. It would be necessary to add other evaluations to know if there are no important associations or if the test is deficient. Also, they should explain how they determined that the participants had a certain degree of body positivity, or if they were informed about the research when they were recruited, which would affect their level of positivity. Likewise, it would be important to relate the variables of fitness in women that may interact with these eating or psychological behaviors.
Author Response
Thank you so much for taking the time to evaluate our work. We have tried to incorporate all your valuable suggestions. If we could improve our work in any way, please let us know.
Comments 1
The aim in the abstract and introduction is broad and sometimes shifts between “opinions about the movement” and “impact of body image.” It should be clarified that the study focuses on opinions toward the movement rather than self-reported body image, as this distinction affects interpretation.
Response: Thank you very much for your suggestion. We have made appropriate changes to the abstract and introduction to make it clear that the study concerned opinions on the body positivity movement, not self-assessment of appearance.
Comments 2
Consider explicitly stating that causality cannot be inferred due to the cross-sectional design.
Response: Thank you for pointing this out. We have added an explicit statement that due to the cross-sectional nature of the project, causal inferences cannot be made.
Comments 3
The body positivity opinion questionnaire is yours self-developed. Although Cronbach’s α is reported, no further psychometric validation is described. This limits the interpretability of results. A brief justification for using this tool and discussion of its limitations should be added to the Methods and Limitations.
Response: Thank you very much for your suggestion. We expanded the description in the Methods section, adding justification for the use of the proprietary tool and a discussion of its limitations. We emphasized that although we obtained a satisfactory Cronbach's α coefficient (0.86), the lack of full psychometric validation limits the interpretation of the results.
Comments 4
Several trends (e.g., higher emotional and uncontrolled eating in BMI ≥25) are described as “may suggest” potential associations. These interpretations should be presented cautiously to avoid overstating non-significant findings.
Response: Thank you very much for your suggestion. We have modified the passages. We now present them more cautiously, emphasizing that they did not reach statistical significance.
Comments 5
The regression R² values (0.106 for BMI, 0.072 for age) indicate limited explanatory power. This should be acknowledged in the discussion to temper conclusions.
Response: We agree with this comment. We have supplemented the paper with a note that the R² values (for BMI and age) indicate the limited explanatory power of the models, which should be taken into account when interpreting the results.
Comments 6
Some of the discussion on media portrayal of body positivity is speculative. Where possible, link these interpretations to existing empirical findings to strengthen the argument.
Response: Thank you very much for your suggestion. We have rewritten the part of the discussion relating to the role of the media. We have tried to base our interpretations on existing empirical reports and have limited speculative elements.
Comments 7
Minor grammatical issues are present (e.g., “idea body positivity” should be “the idea of body positivity,” “loss weight” should be “weight loss”).
Response: We thank you for noting grammatical issues. We have made every effort to improve the overall quality of English in the manuscript to ensure clarity and readability.
Comments 8
Some sentences are long and could be split for clarity, especially in the Discussion.
Response:Thank you for this suggestion. The text has been rewritten to improve clarity and readability.
Comments 9
Check for duplicates (e.g., Löffler et al. appears twice with similar data).
Response:Thank you for this suggestion. The duplicate citation of Löffler et al. has been corrected, and the list of references has been expanded.
Revised the manuxcript to meet expectations.
Thank you for your help. Your guidance is invaluable.
Kind regards,
Authors
Reviewer 2 Report
Comments and Suggestions for Authors
Dear authors,
The manuscript addresses an important and timely topic by examining the relationship between attitudes toward the body positivity movement, eating behaviors, and BMI in women attending fitness classes. The study is relevant for public health, psychology, and nutrition science, and the results could inform interventions targeting self-acceptance and healthy lifestyle promotion. The article is well-structured, follows a logical progression from introduction to conclusions, and uses validated measurement tools. Here are recommendations:
-
- The aim in the abstract and introduction is broad and sometimes shifts between “opinions about the movement” and “impact of body image.” It should be clarified that the study focuses on opinions toward the movement rather than self-reported body image, as this distinction affects interpretation.
- Consider explicitly stating that causality cannot be inferred due to the cross-sectional design.
- The body positivity opinion questionnaire is yours self-developed. Although Cronbach’s α is reported, no further psychometric validation is described. This limits the interpretability of results. A brief justification for using this tool and discussion of its limitations should be added to the Methods and Limitations.
- Several trends (e.g., higher emotional and uncontrolled eating in BMI ≥25) are described as “may suggest” potential associations. These interpretations should be presented cautiously to avoid overstating non-significant findings.
- The regression R² values (0.106 for BMI, 0.072 for age) indicate limited explanatory power. This should be acknowledged in the discussion to temper conclusions.
- Some of the discussion on media portrayal of body positivity is speculative. Where possible, link these interpretations to existing empirical findings to strengthen the argument.
- Minor grammatical issues are present (e.g., “idea body positivity” should be “the idea of body positivity,” “loss weight” should be “weight loss”).
- Some sentences are long and could be split for clarity, especially in the Discussion.
- Check for duplicates (e.g., Löffler et al. appears twice with similar data).
Best regards,
Author Response

(The authors gave the same response as above.)

Reviewer 3 Report
Comments and Suggestions for Authors
Even though the manuscript might be impressive, there are many important points that need clarification, refinement, rewriting, and more information to improve this article.
- The manuscript needs writing and language editing. Please remove the strike through. There is no need to write it in the abstract. Authors should not use the words that appear in the title as keywords. The main aim must be direct and the same throughout the manuscript (abstract, introduction, results/discussion. References should be recent and relevant, they should be well referenced, and their use should be improved throughout the manuscript.
- The introduction section should improve. Line 42: Incomplete idea. In what context is the development of obesity the biggest problem? The introduction section is too long. It would be better to summarize all the topics in three paragraphs. An adequate presentation and a clear justification for conducting this study should be provided. It would be advisable for the authors to present a hypothesis before the primary objective. Why do the authors believe this study is important, and what question would it answer?
- The material and methods section require improvements. What type of study was this? The description should be clear, concise, and detailed. In this section, the authors should describe all procedures performed, variables studied, and statistical analyses performed. Delete any outcomes described here: lines 106-108. State the period or duration of this study. What method (television, flyers, telephone, etc.) was used to recruit participants? It would be best to specify the inclusion (healthy women) and exclusion criteria; for example, did the authors exclude pregnant women, severely obese women, and chronically ill patients. What sociodemographic data did the authors collect? Write the ethical considerations as a subsection. How did participants agree to participate (informed consent)? Lines 114-115: How many questions? Lines 126-128: It would be a good idea to describe how these scales were scored. Line 137, 216: Only the first time an abbreviation appears, the full name should be written. Line 144: So, are the authors referring to patients with overweight and obesity? All variables evaluated in this study must be described, defined, and measured appropriately. This section must provide enough detail on the study design for it to be replicable.
- The results section should be improved. In the text, the authors should write the most significant results, and they should avoid repeating the same information in the text if this data appears in the tables or figures, such as p-values. Here, describe the number of participants and their sociodemographic characteristics. All information to be collected must be described in the M&M section. Table 1: These features should be described in the M&M section. Please write the results with a single decimal place. Lines 155-156: When recruiting participants, the authors should clearly describe the conditions under which the women participated, whether they were healthy. Both the lowest BMI of 16 kg/m² and the highest of 41.9 kg/m² are within the range of severe malnutrition and morbid obesity, which could skew the results and their meaning. Consider re-evaluating the statistics. Lines 168-169: It would be a good idea to add a table with the questions and the score in the M&M section. Line 179-180, 182-183, 192-195, 204: Delete it. The significance of these results should be described and analyzed in the discussion section. Lines 188-190: This interrelationship should be described in the M&M section. Line 190: between BMI and body positivity... Line 201: This statistical analysis should be described in the M&M section. Lines 207-208: This description should appear in the M&M section. It should be clear which were the most significant results of this study. The quality of the tables should be improved.
- The discussion section should be improved, made more argumentative and critical. It should start with the main objective of this study and the most significant results found. The results must be discussed from multiple angles and placed in context without being over-interpreted. Lines 222-225: The main objective should be straightforward and consistent throughout the manuscript. Avoid repeating similar information (Lines 215-207). Authors should indicate why their results are important, how they contribute to current knowledge on this topic, and how they might be applied in clinical practice. What would be the implications of these results?
- Conclusions: What the authors write in the conclusion is part of the discussion section. The introduction, the study design, and the discussion of the results should lead the reader to the same conclusion as the authors.
I would like to encourage the authors to rewrite this manuscript, thinking about the main objective of this study, its design and responding with the results and arguments of the discussion to the most appropriate conclusion of this research work.

The English could be improved to more clearly express the research.
Author Response
Thank you so much for taking the time to evaluate our work. We have tried to incorporate all your valuable suggestions. If we could improve our work in any way, please let us know.
Comments 1
The manuscript needs writing and language editing. Please remove the strike through. There is no need to write it in the abstract. Authors should not use the words that appear in the title as keywords. The main aim must be direct and the same throughout the manuscript (abstract, introduction, results/discussion. References should be recent and relevant, they should be well referenced, and their use should be improved throughout the manuscript.
Response: We appreciate your attention. We have made every effort to improve the manuscript stylistically and linguistically, improving its readability. The abstract has been revised. The main purpose has been clearly and coherently presented in the abstract, introduction, results, and discussion. Keywords have been revised. The bibliography has been updated to include the most recent and relevant sources, and its use has been improved throughout the manuscript.
Comments 2
The introduction section should improve. Line 42: Incomplete idea. In what context is the development of obesity the biggest problem? The introduction section is too long. It would be better to summarize all the topics in three paragraphs. An adequate presentation and a clear justification for conducting this study should be provided. It would be advisable for the authors to present a hypothesis before the primary objective. Why do the authors believe this study is important, and what question would it answer?
Response: Thank you for your comments on the introduction. The section has been revised and shortened to three concise paragraphs. An incomplete sentence in line 42 has been corrected. A clear justification for conducting the study and a hypothesis have been added before the main objective. We have added the study period and a description of the participant recruitment method. We have specified the inclusion criteria (healthy, physically active women) and exclusion criteria (pregnancy, chronic diseases, severe obesity preventing participation in classes).
Comments 3
The material and methods section require improvements. What type of study was this? The description should be clear, concise, and detailed. In this section, the authors should describe all procedures performed, variables studied, and statistical analyses performed. Delete any outcomes described here: lines 106-108. State the period or duration of this study. What method (television, flyers, telephone, etc.) was used to recruit participants? It would be best to specify the inclusion (healthy women) and exclusion criteria; for example, did the authors exclude pregnant women, severely obese women, and chronically ill patients. What sociodemographic data did the authors collect? Write the ethical considerations as a subsection. How did participants agree to participate (informed consent)? Lines 114-115: How many questions? Lines 126-128: It would be a good idea to describe how these scales were scored. Line 137, 216: Only the first time an abbreviation appears, the full name should be written. Line 144: So, are the authors referring to patients with overweight and obesity? All variables evaluated in this study must be described, defined, and measured appropriately. This section must provide enough detail on the study design for it to be replicable.
Response: We appreciate your valuable suggestions. The "Materials and Methods" section has been revised. We clearly define the study type and provide a concise description of all procedures, variables, and statistical analyses. The study period and recruitment method have been specified. Inclusion and exclusion criteria have been described (e.g., exclusion of pregnant women and those with severe obesity or chronic diseases). Outliers in BMI (16 and 41.9) have been removed, and statistical calculations have been recalculated. We have moved all methodological information from the Results section to the Methods section. We have clearly described the number of participants. The process for obtaining informed consent has been specified. This section now includes sufficient detail to ensure reproducibility. Data in tables have been edited to one decimal place.
We thank the reviewer for pointing out overlapping remarks with Comment 3. All issues mentioned have been addressed in the revised.
Comments 4
The results section should be improved. In the text, the authors should write the most significant results, and they should avoid repeating the same information in the text if this data appears in the tables or figures, such as p-values. Here, describe the number of participants and their sociodemographic characteristics. All information to be collected must be described in the M&M section. Table 1: These features should be described in the M&M section. Please write the results with a single decimal place. Lines 155-156: When recruiting participants, the authors should clearly describe the conditions under which the women participated, whether they were healthy. Both the lowest BMI of 16 kg/m² and the highest of 41.9 kg/m² are within the range of severe malnutrition and morbid obesity, which could skew the results and their meaning. Consider re-evaluating the statistics. Lines 168-169: It would be a good idea to add a table with the questions and the score in the M&M section. Line 179-180, 182-183, 192-195, 204: Delete it. The significance of these results should be described and analyzed in the discussion section. Lines 188-190: This interrelationship should be described in the M&M section. Line 190: between BMI and body positivity... Line 201: This statistical analysis should be described in the M&M section. Lines 207-208: This description should appear in the M&M section. It should be clear which were the most significant results of this study. The quality of the tables should be improved.
Response: We appreciate your constructive suggestions. The "Results" section has been thoroughly revised. We are now focusing on the most important findings. The number of participants has been described, and all relevant details have been appropriately moved to the "Materials and Methods" section. Results are consistently reported to one decimal place. The recruitment conditions and health status of participants have been clarified, and the statistical analyses have been carefully reassessed. Text that was incorrectly placed in the "Results" section (lines 179–180, 182–183, 192–195, 204) has been removed, and relevant information has been moved to the appropriate locations. The interrelationships between variables and the description of the statistical analyses have also been moved to the "Materials and Methods" section.
Comments 5
The discussion section should be improved, made more argumentative and critical. It should start with the main objective of this study and the most significant results found. The results must be discussed from multiple angles and placed in context without being over-interpreted. Lines 222-225: The main objective should be straightforward and consistent throughout the manuscript. Avoid repeating similar information (Lines 215-207). Authors should indicate why their results are important, how they contribute to current knowledge on this topic, and how they might be applied in clinical practice. What would be the implications of these results?
Response: The "Discussion" section has been rewritten to enhance clarity and critical analysis. The results are placed in the context of the recent literature, avoiding overinterpretation. Duplicates have been removed. The significance and contribution of the study, as well as potential clinical implications and applications, have been emphasized. We have placed the results in a broader context, indicating their theoretical and practical significance, as well as their limitations in interpretation.
Comments 6
Conclusions: What the authors write in the conclusion is part of the discussion section. The introduction, the study design, and the discussion of the results should lead the reader to the same conclusion as the authors.
Response: The Conclusions have been revised to clearly follow from the introduction, study design, and discussion. The final section now presents a concise synthesis of the study’s findings, aligned with the main objective, without repeating elements of the discussion.
In response to the comments, we have thoroughly revised the language and structure of the manuscript, clarified the objective and hypothesis, detailed the methods, reorganized the results, strengthened the discussion, and rewritten the conclusions. We hope that the revised version of the article presents our research in a much more coherent and transparent manner.
Kind regards,
Authors
Reviewer 4 Report
Comments and Suggestions for Authors
I really appreciated the topic of the article. The relationship between body positivity, body image, and eating behaviours is undoubtedly fascinating and affects many social groups.
That said, there are a few things that could be improved.
On line 107, you repeat twice that the women are gym members. However, you do not specify how often they exercise, or whether there is a significant difference between the amount and type of exercise performed.
- In line 115, you write: 'TFEQ-13 in its Polish adaptation'. What does this mean? Has it been translated into Polish, or have you adapted it for the local culture?
The comparison in lines 232–233 does not seem appropriate to me as the Chinese conception/perception of the body is very different from that in Europe or Australia.
It would be interesting to:
- Expand the sample of women who do not attend gyms to strengthen generalisability.
- Include psychological variables such as anxiety, depression and self-esteem, as well as socio-economic variables, as potential confounders.
Also, include concepts such as body neutrality and fitspiration.
Author Response
We sincerely thank you for your valuable and constructive comments, which have helped us improve the clarity and quality of our manuscript. Please find below our detailed responses to each point:
Comment 1
On line 107, you repeat twice that the women are gym members. However, you do not specify how often they exercise, or whether there is a significant difference between the amount and type of exercise performed.
Response: Thank you for noticing this repetition. We have corrected the wording to avoid redundancy. Additionally, we have now included information on the frequency and type of exercise performed by the participants, which adds more context to their level of physical activity.
Comment 2
In line 115, you write: 'TFEQ-13 in its Polish adaptation'. What does this mean? Has it been translated into Polish, or have you adapted it for the local culture?
Response: We agree that the original statement was unclear. We have revised this section to clarify that we used a validated Polish translation of the TFEQ-13 questionnaire (citing Dzielska et al., 2009; Brytek-Matera et al., 2017). This means that the tool was linguistically translated and psychometrically validated for the Polish population, rather than culturally adapted by us for the purposes of this study.
Comment 3
The comparison in lines 232–233 does not seem appropriate to me as the Chinese conception/perception of the body is very different from that in Europe or Australia.
Response: We acknowledge that cross-cultural comparisons should be made cautiously. We have modified this section to emphasize the cultural differences between body image perceptions.
Comment 4
Expand the sample of women who do not attend gyms to strengthen generalisability.
Response: We fully agree that including women who do not attend gyms would strengthen the generalisability of the findings. Unfortunately, our current study design and resources did not allow us to recruit such a group.
Comment 5
This is a very valuable suggestion. Although our study did not include variables such as anxiety, depression, self-esteem, or socioeconomic status, we recognize their potential impact on eating behaviors and body image. We believe that these factors should be addressed in future research.
Response: Include psychological variables such as anxiety, depression and self-esteem, as well as socio-economic variables, as potential confounders.
Comment 6
Also, include concepts such as body neutrality and fitspiration.
Response: Thank you for pointing out these important and emerging concepts. We have added a short discussion of body neutrality and fitspiration, noting how they intersect with body positivity and may influence body image and health behaviours differently.
We hope that these revisions and clarifications adequately address your comments and improve the quality of our manuscript.
Kind regards,
Authors
Round 2
Reviewer 1 Report
Comments and Suggestions for Authors
The authors have made the requested changes and the text now has better coherence and clarity.
Author Response
We sincerely thank the Reviewer for the additional evaluation of our revised manuscript and for acknowledging the improvements made. We truly appreciate your positive feedback regarding the enhanced coherence and clarity of the text.
Minor stylistic and formatting adjustments have also been made to further improve readability and alignment with journal standards.
We are grateful for your constructive comments throughout the review process, which have significantly contributed to the overall quality and scientific rigor of our paper.
Thank you once again for your time and valuable insights.
Kind regards,
Authors
Reviewer 3 Report
Comments and Suggestions for Authors
The manuscript still presents many important points that require clarification, refinement, rewriting, and further information before it can be published.
Abstract: Please remove the strikethrough. There is no need to write it here.
Introduction: Line 208: body mass index (BMI). Lines 217-388: In the context of the impact of stigma on eating behaviors, it is critical to consider whether the body positivity movement influences eating patterns. That is, does greater body self-acceptance promote healthier eating habits? The current literature does not offer a clear answer to this question. Based on previous research indicating that the body positivity movement can affect body image, self-acceptance, and psychological well-being, albeit with limited evidence on its relationship with eating habits and BMI, we hypothesized whether women with a higher BMI (≥25) would express more critical opinions about the body positivity movement and report higher levels of emotional eating and cognitive restraint compared to women with a normal BMI. Therefore, to address this deficiency, the present study evaluated the opinions of women participating in fitness classes about the body positivity movement and the relationship between these opinions, eating habits, and BMI. //Since existing psychometric tools did not allow for a comprehensive assessment of attitudes toward body positivity in relation to eating behaviors and body mass indices, we developed an original questionnaire for this cross-sectional study that would allow for the simultaneous analysis of participants' opinions, eating behaviors, and BMI scores at a specific time point. The results would provide a better understanding of whether promoting self-acceptance through body positivity contributes to greater well-being and a healthy lifestyle, or whether, under certain circumstances, it can limit motivation to change pro-health behaviors.
Material and Methods: Line 391: The cross-sectional study. Lines 402-403, 795-798: Avoid repeating the same information written previously. Lines 405-408, 414-421, Table 1: This information is part of the results section. Line 408: Women from what age. Line 808: If the authors developed this questionnaire, it would be helpful to display the questions in a table. Was this questionnaire validated? Line 823: So are the authors referring to patients with overweight and obesity?
Results: Line 831: The number of participants, their sociodemographic characteristics, and Table 1 (what was crossed out in M&M) are described below. Table 3 is incomprehensible.
Discussion: It should start with the main objective of this study and the most significant results found. It would be a good idea to divide the discussion based on the subsections presented in the results.
Please also refer to the comments in the document

The English could be improved to more clearly express the research.
Author Response
We sincerely thank the Reviewer for taking the time to carefully read and evaluate our manuscript. We truly appreciate all the constructive comments and suggestions, which have greatly contributed to improving the quality and clarity of our paper. Below, we provide a detailed, point-by-point response to each comment.
Comments 1
Abstract: Please remove the strikethrough. There is no need to write it here.
Response: Thank you very much for your suggestion. The strikethrough has been removed from the abstract. The section has been carefully reviewed to ensure consistency and clarity.
Comments 2
Introduction: Line 208: body mass index (BMI).
Response: Thank you very much for your suggestion. We have made appropriate changes.
Lines 217-388: In the context of the impact of stigma on eating behaviors...
Response: Thank you very much for your suggestion. This part of the introduction has been revised for clarity and conciseness. Repetitive and non-essential content was removed to maintain focus on the study rationale and objectives.
Comments 3
Material and Methods: Line 391: The cross-sectional study.
Response: We agree with this comment. Corrected to: “The cross-sectional study
Lines 402-403, 795-798: Avoid repeating the same information written previously.
Response: Thank you very much for your suggestion. The redundant information was removed.
Lines 405-408, 414-421, Table 1: This information is part of the results section.
Response: These elements were moved from the Materials and Methods section to the Results section, in accordance with the Reviewer’s recommendation.
Line 408: Women from what age.
Response: We agree with this comment. The inclusion criterion has been clarified.
Line 808: If the authors developed this questionnaire, it would be helpful to display the questions in a table. Was this questionnaire validated?
Response: We agree with this comment. The full list of questions from the original body positivity questionnaire has been added in a new table (Table 1). Cronbach’s α coefficient was reported to demonstrate satisfactory internal consistency. The lack of full psychometric validation was acknowledged in the Materials and Methods section and discussed as a limitation.
Line 823: So are the authors referring to patients with overweight and obesity?
Response :This has been clarified.
Comments 4
Results: Line 831: The number of participants, their sociodemographic characteristics, and Table 1 (what was crossed out in M&M) are described below. Table 3 is incomprehensible.
Response: Thank you for this suggestion. Corrections were made.
Table 3 is incomprehensible
Response :Table 3 has been reformatted for improved readability and clarity. Column labels and legends have been improved to present the data more clearly. The manuscript has been submitted in a non-editable format to avoid distortions in the table.
Comments 5
Discussion: It should start with the main objective of this study and the most significant results found. It would be a good idea to divide the discussion based on the subsections presented in the results.
Response: Thank you very much for your suggestion. We have made the modifications.
Comments 6
Please also refer to the comments in the document
Response: Thank you for this suggestion. All additional in-text comments and editorial notes have been addressed in the revised version of the manuscript. Minor grammatical and stylistic corrections were made throughout the text to enhance precision and fluency.
Revised the manuxcript to meet expectations.
We would like to once again thank the Reviewer for the valuable feedback and insightful remarks that helped us significantly improve the manuscript. Your guidance is invaluable.
Kind regards,
Authors

Round 3
Reviewer 3 Report
Comments and Suggestions for Authors
Even though the manuscript has improved, there are some key questions the authors must answer before its publication:
Line 96: This qualitative and cross-sectional study...
Line 100: The authors must describe the method they used (fitness website, television, brochures, telephone, etc.) to recruit the participants. How did the authors calculate the sample size to obtain significant results?
Lines 120-121: It would be better to write TFEQ-13 instead of the full name. The TFEQ-13 is a validated Polish version that assesses three dimensions of eating behaviors.
Line 124: TFEQ-13...
Line 151: it would be a clever idea to write “overweight with BMI < 25 kg/m² versus obesity with BMI ≥ 25 kg/m²”.
Line 160: It good be better to write “A total of 118 women between the ages of 18 and 65 participated in the study” and delete what is crossed out (lines 163-164).
Line 165: How many participants were BMI < 25 kg/m² and BMI ≥ 25 kg/m?
Line 188: Write the full name: Body mass index..
Lines 190-191: the authors need to be more specific, for example, “In all participants, the mean score on the cognitive restriction subscale was higher than on the uncontrolled eating and emotional eating subscales.”
Line 192, 193, 194, 200, 201, 202-203, 207, 208, 210, 211: Avoid repeating the same information in the text if this data appears in the tables. Remove anything that is crossed out.
Line 209: Avoid writing comments like "However." In the results section, authors should state their results without any explanation.
Discussion: It would be more appealing to divide this section into subsections to focus on important topics developed in this study, for example, Body acceptance (Lines 228-251), Age and eating behaviors (Line 252-266), Media messages and cultural context (Lines 267-310).
Lines 324-326: No human study that has not received approval from an ethics committee would be approved for execution, much less publish its results; therefore, it is not necessary to write this paragraph.
Lines 340-342: Why was this original questionnaire not validated with the information obtained from the 118 participants?
Lines 343-363: These two paragraphs should be written before "Strengths and Limitations".

The English could be improved to more clearly express the research.
Author Response
Thank you so much for taking the time to evaluate our work. We have tried to incorporate all your valuable suggestions. If we could improve our work in any way, please let us know.
- Comments 1: Line 96:This qualitative and cross-sectional study...
Response: The sentence has been corrected as suggested.
- Comments 2: Line 100:The authors must describe the method they used (fitness website, television, brochures, telephone, etc.) to recruit the participants. How did the authors calculate the sample size to obtain significant results?
Response: We have added a detailed description of the recruitment method. Participants were recruited through posts published on gym websites and social media platforms.
- Comments 3: Lines 120-121: It would be better to write TFEQ-13 instead of the full name. The TFEQ-13 is a validated Polish version that assesses three dimensions of eating behaviors.
Response: We agree with this suggestion. The name has been shortened to TFEQ-13, with an explanation provided upon first use.
- Comments 4: Line 124: TFEQ-13...
Response: We revised the sentence to ensure consistency throughout the manuscript and to correctly refer to the TFEQ-13.
- Comments 5: Line 151: it would be a clever idea to write “overweight with BMI < 25 kg/m² versus obesity with BMI ≥ 25 kg/m²”.
Response: We appreciate this suggestion; however, according to WHO classifications, overweight is defined as BMI 25.0–29.9 kg/m² and obesity as BMI ≥ 30.0 kg/m². In our study, the division into groups was based on a different criterion (BMI < 25 kg/m² and BMI ≥ 25 kg/m²), as this cut-off point reflects the distinction between normal weight and excess body weight rather than between overweight and obesity.
- Comments 6: Line 160: It good be better to write “A total of 118 women between the ages of 18 and 65 participated in the study” and delete what is crossed out (lines 163-164).
Response: The sentence has been corrected exactly as suggested, and unnecessary text has been removed.
- Comments 7: Line 165: How many participants were BMI < 25 kg/m² and BMI ≥ 25 kg/m?
Response: This information has been added: 67 participants had a BMI < 25 kg/m², and 51 had a BMI ≥ 25 kg/m².
- Comments 8: Line 188: Write the full name: Body mass index..
Response: The full term “Body Mass Index” has been added.
- Comments 9: Lines 190-191: the authors need to be more specific, for example, “In all participants, the mean score on the cognitive restriction subscale was higher than on the uncontrolled eating and emotional eating subscales.”
Response: The sentence has been corrected as suggested.
- Comments 10: Line 192, 193, 194, 200, 201, 202-203, 207, 208, 210, 211: Avoid repeating the same information in the text if this data appears in the tables. Remove anything that is crossed out.
- Line 209: Avoid writing comments like "However." In the results section, authors should state their results without any explanation.
Response: All strikethroughs removed. The results section now presents findings concisely and without interpretative comments.
- Comments 11: Discussion: It would be more appealing to divide this section into subsections to focus on important topics developed in this study, for example, Body acceptance (Lines 228-251), Age and eating behaviors (Line 252-266), Media messages and cultural context (Lines 267-310).
Response: The Discussion has been reorganized into subsections as suggested. The new subsections include:
- Body acceptance
- Age and eating behaviors
- Media messages and cultural context
- Comments 12: Lines 324-326: No human study that has not received approval from an ethics committee would be approved for execution, much less publish its results; therefore, it is not necessary to write this paragraph.
Response: This paragraph has been removed as recommended.
- Comments 13: Lines 340-342:Why was this original questionnaire not validated with the information obtained from the 118 participants?
Response: The suggested changes have been implemented.
- Comments 14: Lines 343-363: These two paragraphs should be written before "Strengths and Limitations".
Response: The suggested changes have been implemented.
We hope that our revisions satisfactorily address all the Reviewer's comments. We remain grateful for the valuable feedback and are open to making any additional clarifications or improvements if required.
Authors